# Exploring the Pathophysiologic Cascade Leading to Osteoclastogenic Activation in Gaucher Disease Monocytes Generated via CRISPR/Cas9 Technology

**DOI:** 10.3390/ijms241311204

**Published:** 2023-07-07

**Authors:** Maximiliano Emanuel Ormazabal, Eleonora Pavan, Emilio Vaena, Dania Ferino, Jessica Biasizzo, Juan Marcos Mucci, Fabrizio Serra, Adriana Cifù, Maurizio Scarpa, Paula Adriana Rozenfeld, Andrea Elena Dardis

**Affiliations:** 1Regional Coordinator Centre for Rare Diseases, Academic Hospital of Udine, 33100 Udine, Italy; maxi.ormazabal@gmail.com (M.E.O.); pavan.eleonora@gmail.com (E.P.); fabrizio.serra@asufc.sanita.fvg.it (F.S.); maurizio.scarpa@asufc.sanita.fvg.it (M.S.); 2Instituto de Estudios Inmunológicos y Fisiopatológicos (IIFP), UNLP, CONICET, Asociado CIC PBA, Facultad de Ciencias Exactas, Departamento de Ciencias Biológicas, La Plata 1900, Argentina; emiliovaena@gmail.com (E.V.); juanmarcosmucci@gmail.com (J.M.M.); paularozenfeld@gmail.com (P.A.R.); 3Institute of Clinical Pathology, Department of Laboratory Medicine, University Hospital of Udine, 33100 Udine, Italy; dania.ferino@asufc.sanita.fvg.it (D.F.); jessica.biasizzo@asufc.sanita.fvg.it (J.B.); 4Dipartimento di Area Medica, Università degli Studi di Udine, 33100 Udine, Italy; adriana.cifu@uniud.it

**Keywords:** Gaucher disease, bone, monocytes, osteoclasts, inflammation

## Abstract

Gaucher disease (GD) is caused by biallelic pathogenic variants in the acid β-glucosidase gene (*GBA1*), leading to a deficiency in the β-glucocerebrosidase (GCase) enzyme activity resulting in the intracellular accumulation of sphingolipids. Skeletal alterations are one of the most disabling features in GD patients. Although both defective bone formation and increased bone resorption due to osteoblast and osteoclast dysfunction contribute to GD bone pathology, the molecular bases are not fully understood, and bone disease is not completely resolved with currently available specific therapies. For this reason, using editing technology, our group has developed a reliable, isogenic, and easy-to-handle cellular model of GD monocytes (GBAKO-THP1) to facilitate GD pathophysiology studies and high-throughput drug screenings. In this work, we further characterized the model showing an increase in proinflammatory cytokines (Interleukin-1β and Tumor Necrosis Factor-α) release and activation of osteoclastogenesis. Furthermore, our data suggest that GD monocytes would display an increased osteoclastogenic potential, independent of their interaction with the GD microenvironment or other GD cells. Both proinflammatory cytokine production and osteoclastogenesis were restored at least, in part, by treating cells with the recombinant human GCase, a substrate synthase inhibitor, a pharmacological chaperone, and an anti-inflammatory compound. Besides confirming that this model would be suitable to perform high-throughput screening of therapeutic molecules that act via different mechanisms and on different phenotypic features, our data provided insights into the pathogenic cascade, leading to osteoclastogenesis exacerbation and its contribution to bone pathology in GD.

## 1. Introduction

Gaucher disease (GD—OMIM #230800), one of the most prevalent lysosomal storage disorders, is caused by the deficient activity of glucocerebrosidase enzyme (GCase EC 3.2.1.45) due to biallelic pathogenic variants in the gene encoding the acid β-glucocerebrosidase protein, *GBA1* (GRCh37/hg19 Chromosome 1: 155,204,239 to 155,214,653). This enzyme deficiency causes a progressive accumulation of glucosylceramide (GlcCer) and its deacylated metabolite glucosylsphingosine (GlcSph), mainly in the monocyte/macrophage system [1]. Lipid-laden macrophages or “Gaucher cells” infiltrate the bone marrow, spleen, and liver, representing GD’s hallmark [2]. Gaucher cells produce a peculiar signature of inflammatory genes [3].

Historically, this disease has been classified into three clinical phenotypes based on the presence or absence and severity of neurological symptoms. The most frequent clinical form of GD is type 1 (OMIM # 230800), present in about 90% of patients. Type 1 GD is characterized by the enlargement of the liver and spleen and the displacement of normal bone marrow by storage cells, anemia, thrombocytopenia, bleeding, and skeletal alterations [1]. Although type 1 GD is considered a non-neuronopathic form, there is increasing evidence that neurological involvement manifested as Parkinson’s syndrome can also occur [4]. Bone manifestations in GD include bone infarcts, avascular bone necrosis, lytic lesions, osteosclerosis, fractures due to osteopenia or osteoporosis, and rarely acute osteomyelitis. Bone pain of varying intensity, fractures, and progressive joint collapses may cause impaired mobility and increased morbidity [5]. To date, the management of bone pathology remains challenging even after the introduction of specific disease-modifying therapies such as enzyme replacement therapy (ERT: imiglucerase, velaglucerase alfa, and taliglucerase) or substrate reduction therapy (SRT: eliglustat and miglustat) [6], which have a high economic impact on the health system [7,8,9].

Physiological bone remodeling is a highly coordinated process that requires the concerted function of osteoclasts and osteoblasts involved in bone resorption and bone formation, respectively. This process depends on the orchestration of multiple cell types and relies on the tightly regulated expression of specific genes [10,11,12,13]. Bone remodeling is affected in GD. However, the molecular mechanisms responsible for the clinical manifestations of bone disease in GD are not completely understood. Several lines of evidence indicate that both increased osteoclast-mediated bone resorption, as well as decreased osteoblast-mediated bone formation, would be involved. Furthermore, osteocytes are also affected in GD [14,15,16].

Chronic stimulation of the immune system is a hallmark of many lysosomal disorders, including GD, and it is characterized by a markedly elevated level of proinflammatory cytokines, such as Interleukin-1β (IL-1β), Tumor Necrosis Factor- α (TNF-α), and/or Interleukin-6 (IL-6), and chemokines [17]. A possible pathogenic role of the immune system in bone pathology has been proposed since it has been demonstrated that inflammation may have a negative impact on bone homeostasis by targeting osteoclasts, osteoblasts, and osteocytes [18]. Indeed, two cytokines, macrophage colony-stimulating factor (M-CSF) and receptor activator of nuclear factor κB ligand (RANKL), are absolutely required for osteoclast differentiation. Moreover, other cytokines could also induce osteoclast differentiation under inflammatory pathological conditions [19]. In vitro studies using GD models demonstrated that GCase deficiency leads to increased secretion of proinflammatory cytokines, including TNF-α, associated with an increased number of mature osteoclasts [20]. In addition, enhanced osteoclastogenesis was described due to a synergistic effect of IL-1β and RANKL secreted via the mesenchymal stem cells (MSCs) derived from GD patients [21].

Finally, bone involvement cannot be completely reversed with the current therapies available for GD, and the effect of ERT on proinflammatory state changes has been poorly described [22,23].

Studies to unravel the pathogenic mechanisms of GD have been performed by exploiting the differentiation of patients’ derived iPSCs to specific cell types [24,25,26,27,28], including macrophages [29,30]. However, patient-derived cells are not always easily accessible, and on the other hand, iPSC-derived macrophages are difficult, expensive, and time consuming to generate, culture, and differentiate [29,30,31]. Several studies have used macrophages differentiated from peripheral blood monocytes (PBMCs) [29,32].

Our group has previously exploited the genome editing technology CRISPR/Cas9 to knockout the *GBA1* gene in commercially available continuous cell lines, generating cells that recapitulate the main features of GD. These cell models are isogenic, with control cells easy-to-handle and inexpensive to grow in culture. For these reasons, they represent a useful and convenient in vitro model to investigate GD pathophysiology and a suitable tool for high-throughput drug screening. In particular, a human monocytic cell line (THP-1) *GBA1* knockout was generated (GBAKO-THP1) [33]. This cell line was selected since it can be differentiated into macrophage-like cells [34], the main cell type affected in GD. 

Given the convenience of this model, the aim of this present work is to evaluate the suitability of GBAKO-THP1 to study the osteoclastogenesis and inflammatory profile in GD and to analyze the effectiveness of different therapies in vitro. 

## 2. Results

### 2.1. Characterization of the Monocyte GBA1 Knockout Cells

In previous work, we generated and characterized a *GBA1* knockout monocytic cell line (GBAKO-THP1) as a new cell model of GD [33]. The GBAKO-THP1-generated cells display less than 1% of residual GCase activity and a massive accumulation of glucosylsphingosine (GlcSph). To further validate the GBAKO-THP1 as a model of GD monocytes, we decided to evaluate specific pathological features previously described in monocytes derived from peripheral blood mononuclear cells (PBMC) of GD patients. In particular, we analyzed the GBAKO-THP1 osteoclastogenic potential [35] and the proinflammatory profile since GD PBMC, when cultured in vitro, secreted proinflammatory cytokines and showed a higher tendency to differentiate into functional osteoclasts [14,36].

#### 2.1.1. Increased Osteoclastogenesis in GBAKO-THP1 Cells

The osteoclastogenic potential of the GBAKO-THP1 and wild-type (wt) THP-1 monocytes was assessed by inducing their differentiation to osteoclasts (see Section 4.3 and Figure 1A) and counting the generated osteoclast-like cells. As shown in Figure 1C, the number of osteoclasts (assessed as multinucleated tartrate-resistant acid phosphatase (TRAP)-positive cells/total cells) differentiated from GBAKO-THP1 was significantly higher than the number of osteoclasts differentiated from THP-1 wt cells. This result indicates that, as reported in human GD PBMC [14,36], osteoclastogenesis is increased in GBAKO-THP1 cells. 

#### 2.1.2. Proinflammatory Cytokines Released by GBAKO-THP1 Cells

Considering the interaction between immune and bone systems [18] and the fact that several proinflammatory cytokines, including interleukin-1β (IL-1β), tumor necrosis factor α (TNF-α), and interleukin-6 (IL-6), are elevated in patients with GD and several in vitro disease models [30,37,38,39], we decided to examine the pro-inflammatory profile in monocyte GBAKO-THP1 cells. Therefore, we assessed the levels of IL-1β, TNF-α, and IL-6 secreted by the cells into the culture media. Two of these cytokines, IL-1β and TNF-α, were significantly elevated in GBAKO compared to wt cells (Figure 2). Conversely, the levels of IL-6 released by GBAKO-THP1 cells were not significantly different from those secreted by wt (Appendix A).

#### 2.1.3. Role of Increased IL-1β Levels in Osteoclastogenesis

Osteoclasts’ differentiation from monocytes is stimulated via the essential cytokine receptor activator of nuclear factor κB ligand (RANKL) secreted by MSCs. Osteoimmunological studies revealed that proinflammatory cytokines may affect this process [40]. It was shown that IL-1β participates in osteoclast multinucleation, increases cell survival, and synergizes the effect of RANKL-induced osteoclastogenesis [41]. To evaluate the relative contribution of the increased levels of IL-1β production on the increased osteoclastogenic potential of GBAKO-THP1 monocytes, the osteoclastogenesis experiment (see Figure 1) was carried out in the presence of Anakinra, a specific antagonist of IL-1β receptor. As shown in Figure 3, the number of GBAKO-THP1 -derived osteoclasts was significantly reduced in the presence of Anakinra, suggesting that the increased levels of IL-1β produced via GBAKO-THP1 cells play an important role in the increase in osteoclast differentiation in the context of GD.

### 2.2. Evaluation of the Effect of Treatments on the Phenotype of the GBAKO-THP1

We then evaluated the effect of the following treatments on osteoclastogenesis and inflammatory cytokine profile of GBA1KO-THP1 monocytes: Enzyme replacement (Imiglucerase, recombinant human GCase rhGCase); substrate reduction (Eliglustat, SRT); pharmacological chaperone (Ambroxol, ABX); anti-inflammatory (Pentosan-polysulfate, PPS). In addition, the effects of these compounds on GCase activity and substrate accumulation were measured as well.

#### 2.2.1. Effect on GCase Activity

We tested the effect on GCase enzyme activity of rhGCase and ABX only since, due to their mechanism of action, they were the only treatments that could eventually have an impact on GCase activity. In the case of rhGCase, the recombinant enzyme is supplied and internalized by the cell, while ABX, being a pharmacological chaperone (PC), promotes the correct folding and trafficking to the lysosome of endogenous mutated misfolded but partially active GCase proteins. The GBAKO-THP1 cells express GCase proteins carrying large deletions within the catalytic domain (see above) that most likely completely abrogate the enzyme activity [33]. However, since the mutated GCase protein is expressed, we decided to test the effect of ABX. GBAKO-THP1 cells were incubated with 0.08 UI/mL rhGCase (Imiglucerase) or 100 μM of ABX for 2 or 4 days. As reported in Figure 4, an increase in GCase activity was detected in cells treated with rhGCase at both time points assessed, reaching 60% of the enzyme activity observed in wt cells. On the contrary, treatment with ABX did not exert any effect on enzyme activity.

#### 2.2.2. Effect on Substrate Accumulation

We then decided to evaluate the effect of treatments on intracellular GlcSph levels in GBAKO-THP1 cells treated for 48 and 96 h. GlcSph was significantly reduced not only in rhGCase- and SRT-treated cells as expected but also in cells treated with ABX (Figure 5). The reduction in GlcSph accumulation in the absence of an effect on GCase activity in cells treated with ABX suggests that mechanisms other than the increase in GCase stability and trafficking are involved in the reduction of substrate accumulation. On the contrary, PPS did not affect GlcSph accumulation, suggesting that substrate accumulation is not affected by the inflammatory state. 

#### 2.2.3. Effect of Ambroxol on Substrate Reduction

Based on the results obtained when GBAKO-THP1 cells were treated with ABX, we hypothesized that ABX might induce the extracellular excretion of GlcSph. Indeed, it has been described that the treatment of neuronal cell models with ABX induces exocytosis [42]. For this reason, we decided to assess GlcSph levels in the culture media of ABX-treated GBAKO-THP1 cells. As shown in Figure 6, ABX treatment resulted in significantly increased levels of GlcSph in the extracellular media. Lower levels of GlcSph at 96 h compared to 48 h may be the result of unspecific degradation in extracellular media or instability in suspension. 

#### 2.2.4. Effect on Osteoclastogenesis

We then evaluated the effect of the treatments on osteoclastogenesis in GBAKO-THP1 cells treated throughout the differentiation process. As shown in Figure 7, the number of osteoclasts differentiated from GBAKO-THP1 was significantly reduced by all treatments.

#### 2.2.5. Effect on Proinflammatory Cytokines Release

Finally, we assessed the effect of treatments on the inflammatory profile. As shown in Figure 8, IL-1β release was reduced to the levels observed in wt cells when GBAKO-THP1 monocytes were treated with rhGCase, SRT, or PPS, whereas ABX did not exert any effect on the levels of this cytokine (Figure 8A). The addition of rhGCase, SRT, and ABX to cell cultures of GBAKO-THP1 significantly reduced the secretion of TNF-α (Figure 8B).

## 3. Discussion

Skeletal alterations are one of the most disabling features in GD patients nowadays and are not completely resolved with currently available specific therapies. Both defective bone formation and increased bone resorption due to activated osteoclastogenesis have been demonstrated to contribute to GD bone pathology [15,18,20,21,43]. 

There is a close interaction between the skeletal and immune systems, and it is known that pro-inflammatory cytokines play a key role in the unbalance of bone homeostasis [19,44].

Our laboratory recently generated a human monocyte model of GD by editing the *GBA1* gene in a human monocyte THP-1 cell line. These cells showed abnormal GCase protein expression and activity, along with substrate accumulation as revealed by an increase in glucosylsphingosine (GlcSph) levels, resembling the main features of GD and representing an easy-to-handle, low cost, and isogenic model of GD [33]. In this work, we wanted to validate the suitability of this model to study osteoclastogenesis in GD and analyze in vitro the effect of different available therapies.

To this aim, we first evaluated the capacity of GBAKO-THP1 monocytes to differentiate into osteoclasts. Using the tartrate-resistant acid phosphatase (TRAP) assay to identify osteoclasts, we found that GBAKO-THP1 monocytes display a significantly increased tendency to differentiate into osteoclast-like cells. This result concurs with results obtained using cells derived from GD patients [14,36,45]. 

Considering the close relationship between bone tissue and the immune system [18], we characterized the inflammatory status of GBAKO-THP1, showing a [16] higher production of IL-1β and TNF-α by these cells. 

Moreover, we evaluated the role of IL-1β in the osteoclastogenesis process using the specific inhibitor Anakinra. Our results suggest that the increased levels of IL-1β released via the GBAKO-THP1 contribute, at least in part, to the increased osteogenic potential displayed by these cells.

It is worth underlying that the evidence of increased osteoclastogenesis in GD patients, in comparison with healthy controls reported in the literature, has been obtained by comparing the number of osteoclasts generated via the differentiation of peripheral blood mononuclear cells (PBMCs) [36] or monocytes isolated from the PBMCs pool [14,15]. However, in these models, the increased number of osteoclasts obtained after differentiation may reflect the higher number of pre-osteoclasts identified in the PBMCs population from GD patients, probably due to the enhanced inflammatory environment in which these cells originate [36]. Moreover, monocytes isolated from GD patients’ PBMCs pool have already been interacting with other cells, including T and dendritic cells, which display increased proinflammatory cytokines production in GD patients [36]. Thus, in those systems, the increased osteoclastogenesis cannot be explained only via the GCase deficiency and subsequent lipid accumulation in monocytes themselves. On the contrary, our model consists of naïve cells, which have not been influenced by a microenvironment other than one of the monocytic precursors they derive from.

After validating this characteristic phenotype, we decided to analyze whether this model would be suitable to evaluate different treatment approaches for their ability to restore the pathological phenotype and, at the same time, the contribution of substrate accumulation and inflammation to the increased osteogenic potential of GD monocytes. We evaluated the effect of recombinant human GBA (rhGCase) as an enzyme replacement therapy (ERT); an inhibitor of glucosylceramide synthase (Eliglustat) as a substrate reduction therapy (SRT); a pharmacological chaperone, Ambroxol (ABX); and an anti-inflammatory molecule, pentosan polysulfate (PPS). 

We tested the ability to restore the GCase activity of rhGCase and ABX. rhGCase treatment resulted in a significant increase in GCase activity, while the chaperone ABX had no effect on enzyme activity. This result was expected, considering that although GBAKO-THP1 cells express mutant GCase proteins, they are likely completely inactive. Therefore, even in the case that they could reach the lysosome in the presence of ABX, they would not be able to exert their function. Indeed, according to the genetic variants identified in the *GBA1* gene, these cells would express two protein isoforms lacking residues 40–103 and 40–152, respectively [33]. In accordance with that, the analysis of the *GBA1* mRNA sequence showed that GBAKO-THP1 cells express two transcripts that would be translated into the above-mentioned proteins. In addition, they express a third mRNA variant (not previously identified via DNA analysis) that would be translated into an additional mutant protein lacking residues 52–84 (Appendix A). In all cases, the described deletions would be deleterious for the catalytic activity of GCase since they are located in Domain III containing the catalytic site [46] and are predicted to cause conformational changes in the protein structure (Appendix A). 

We further evaluated the effect of these therapies on intracellular GlcSph levels. ERT and SRT treatments induced a reduction in glycolipid levels, whereas PPS had no effect on the GlcSph accumulation. Surprisingly, although ABX did not induce changes in GCase activity, it reduced the intracellular levels of GlcSph, probably by inducing its release to the extracellular media as suggested by the increased levels of this sphingolipid in the culture media of cells under ABX treatment. The ability of ABX to induce exocytosis has already been observed in primary mouse neurons: ABX treatment induced the secretion of alpha-synuclein by these cells [25]. The induction of exocytosis via ABX treatment is likely due to its properties as a weak base [47].

Finally, we evaluated the effect of these treatments on osteoclastic differentiation and the pro-inflammatory profile. All the treatments were able to restore the osteoclastogenesis of GBAKO-THP1 cells to the levels observed in wt cells. Regarding the proinflammatory profile, all treatments were able to modulate either one or both studied cytokines (IL-1β and TNF-α).

Furthermore, our data suggest that increased osteoclastogenesis in GD would be determined not only via the interaction of monocytes with factors released by other cells, such as MSCs, osteoblasts, etc., but they display an increased intrinsic osteoclastogenic potential which would be independent of their interaction with the GD microenvironment. 

All evaluated treatments were able to ameliorate, at least in part, the pathological phenotype, confirming that this model would be suitable to perform high-throughput screening of molecules that can act via different mechanisms of action and on different phenotypic features.

rhGCase induced, as expected, increased GCase activity with a significant reduction in substrate accumulation, leading to decreased cytokines release and osteoclast differentiation. By treating the cells with a substrate reduction therapy, such as Eliglustat, we confirmed that a reduction in substrate accumulation resulted in a reduction in cytokines release and osteoclastogenesis. Similar results were obtained with ABX that, in this model, acted as an SRT by decreasing GlcSph levels inside the cells. Finally, anti-inflammatory treatment affected only the release of cytokines, which, in turn, resulted in decreased osteoclastogenesis without affecting substrate accumulation. Taken together, these results suggest a sequential model of pathophysiology in GD monocytes in which the increased osteoclastogenesis would be caused by inflammation triggered via intracellular lipid accumulation (Figure 9).

## 4. Materials and Methods

### 4.1. Cell Lines

Human monocytic cells derived from an acute monocytic leukemia patient (THP-1, from ATCC, Manassas, VA, USA) were used in this study as models of human monocytes in their wild-type and GBA knockout (GBA1KO-THP1) versions. GBA1KO-THP1 cells were previously generated in our laboratory using CRISPR/Cas9 editing technology and validated as a model of Gaucher disease [22]. Briefly, GBA1KO-THP1 cells present two large in-frame deletions within the sequence of the *GBA1* gene encoding the catalytic domain of the enzyme (c.115+145_307+1del and c.246_441del). As expected, these cells express a mutated but non-functional GCase protein displaying less than 1% of residual GCase activity. The loss of activity results in a massive accumulation of glucosylsphingosine (GlcSph) within the cells (a 50-fold increase compared to the wt).

### 4.2. Cell Culture

THP-1 wt and GBAKO-THP1 were cultured and maintained in RPMI 1640 medium (EuroClone, Pero, Italy) containing 10% fetal bovine serum (Gibco, Waltham, MA USA), 1% glutamine (Gibco), and 1% penicillin/streptomycin (Gibco) (complete media), in a humidified atmosphere containing 5% CO_2_ at 37 °C.

### 4.3. Osteoclast Differentiation Assay

For direct osteoclastogenesis assays, THP-1 wt and GBAKO-THP1 cells were seeded at 1.5 × 10^5^ cells/well in 24-well plates. THP-1 cells were stimulated for 48 h with 100 ng/mL phorbol-12 myristate-13 acetate (PMA) to induce the cells to differentiate into adherent macrophages. Subsequently, 30 ng/mL of recombinant human macrophage colony-stimulating factor (M-CSF) (PeproTech, Rocky Hill, NJ, USA) and 50 ng/mL of recombinant human sRANK ligand (RANKL) (PeproTech) were added to the medium (complete media) for 7 days replacing the media every 48 h. 

Cells were fixed in 4% paraformaldehyde to identify osteoclasts and stained for tartrate-resistant acid phosphatase (TRAP; Sigma Aldrich, St. Louis, MO, USA) following manufacturer’s instructions or labeled with alpha-phalloidin (Sigma, St. Louis, MO, USA) and Hoechst. Cells were visualized at the microscope Leica DM600B (Leica, Wetzlar, Germany), and osteoclasts were identified as TRAP-positive multinucleated cells (3 or more nuclei), and the number was determined using microscopic counts on a whole well. 

To assess the contribution of IL-1β to osteoclast formation, neutralization experiments were performed using the addition of the antagonist Anakinra (R&D systems, Minnesota, MN, USA) at 500 ng/mL.

### 4.4. Enzymatic Activity

GCase enzymatic activity was measured using the fluorogenic substrate 4-methylumbelliferyl-β-d-glucopyranoside (Sigma-Aldrich, St. Louis, MO, USA). The total amount of protein in cell lysates was determined using the Bradford assay (BioRad, Hercules, CA, USA), following manufacturer’s instructions. A total of 10 µL containing 10 µg of protein were incubated with 10 µL of substrate 5 mM in acetate buffer 0.1 M pH 4.2 at 37 °C for 3 h. The reaction was stopped with carbonate buffer 0.5 M pH 10.7, and the fluorescent product was quantified using a fluorimeter (SPECTRAmax Gemini XPS, Molecular Devices, San Jose, CA, USA) at an excitation wavelength of 365 nm and emission of 495 nm.

### 4.5. Cytokine Measurement

The concentration of interleukin-1β (IL-1β), tumor necrosis factor α (TNF-α), and interleukin-6 (IL-6) in the supernatant fraction were assessed using a microfluidic Simple Plex^TM^ Assay (ProteinSimple, San José, CA, USA) on an ELLA^TM^ instrument. Cell culture supernatant was diluted 1:2 with sample diluent, achieved by adding 35 µL of sample to 35 µL of diluent. Finally, 50 µL of the sample (diluted) was added to the sample well of the cartridge. The instrument was calibrated using the in-cartridge factory standard curve.

### 4.6. Glucosylsphingosine Measurement (GlcSph)

Glucosylsphingosine (GlcSph) was measured using LC-MS/MS technology as previously described [48,49]. D5-glucosylsphingosine was used as an internal standard. Briefly, after protein precipitation, evaporation, and reconstitution in mobile phase, reverse-phase liquid chromatography was performed using a Shimadzu Nexera CL UHPLC (Shimadzu, Kyoto, Japan) and a Poroshell 120 EC-C8 column, 3.0 × 50.0 mm with 2.7 μm particle size (Agilent, Santa Clara, CA, USA). Mass spectrometry detection was carried out with AB Sciex 6500 QTrap tandem mass spectrometer (Sciex, Framingham, MA, USA) set in positive mode using electrospray ionization (ESI). GlcSph levels were normalized by protein content and assessed using TPUC3 total protein urine/CSF (Roche, Basel, Switzerland) in Cobas 8000 (Roche, Basel, Switzerland) following the manufacturer’s instructions.

### 4.7. Cells Treatments

GBAKO THP-1 cells were treated with 1.6 µM recombinant human glucocerebrosidase (rhGCase) imiglucerase as enzyme replacement therapy (Cerezyme; Genzyme, Cambridge, MA, USA). For substrate reduction therapy, cells were treated with 20 µM of D,L-threo-PDMP (Eliglustat—Matreya LLC, State College, PA, USA), a glucosylceramide synthase inhibitor. Ambroxol hydrochloride (A9797, Sigma-Aldrich) and the anti-inflammatory molecule pentosan polysulfate (PPS—Bene pharmaChem, Geretsried, Germany) were used at 100 μM and 5 μg/mL, respectively [45]. In all cases, cells were treated for 48 and 96 h, as previously described [45,48,50].

### 4.8. Statistical Analysis

Analysis was performed using the GraphPad Prism v8 (GraphPad Software, San Diego, CA, USA), applying unpaired *t*-test and ANOVA followed by the Bonferroni test. The normal distribution of data was verified using the Shapiro–Wilk normality test. Data are expressed as mean ± SD and are representative of three independent experiments.

## 5. Limitations and Conclusions

We have shown that the GBAKO-THP1 cells generated in our laboratory using gene editing technology display the main characteristics already described in GD monocytes. However, this model presents some of the following limitations: (1) it is derived from a continuous cell line; (2) it does not represent the phenotypic variability that, in fact, exists among patients affected by the disease. Therefore, the obtained results might not apply to all patients, and confirmatory experiments using patient’s derived cells should be undertaken whenever possible. 

In conclusion, despite this drawback, we have shown that the model presented here recapitulates the main features of GD, including increased proinflammatory cytokines (IL-1β and TNF-α) release and the activation of osteoclastogenesis, and represent a valuable and convenient system to perform high-throughput screening experiments. 

## Figures and Tables

**Figure 1 ijms-24-11204-f001:**
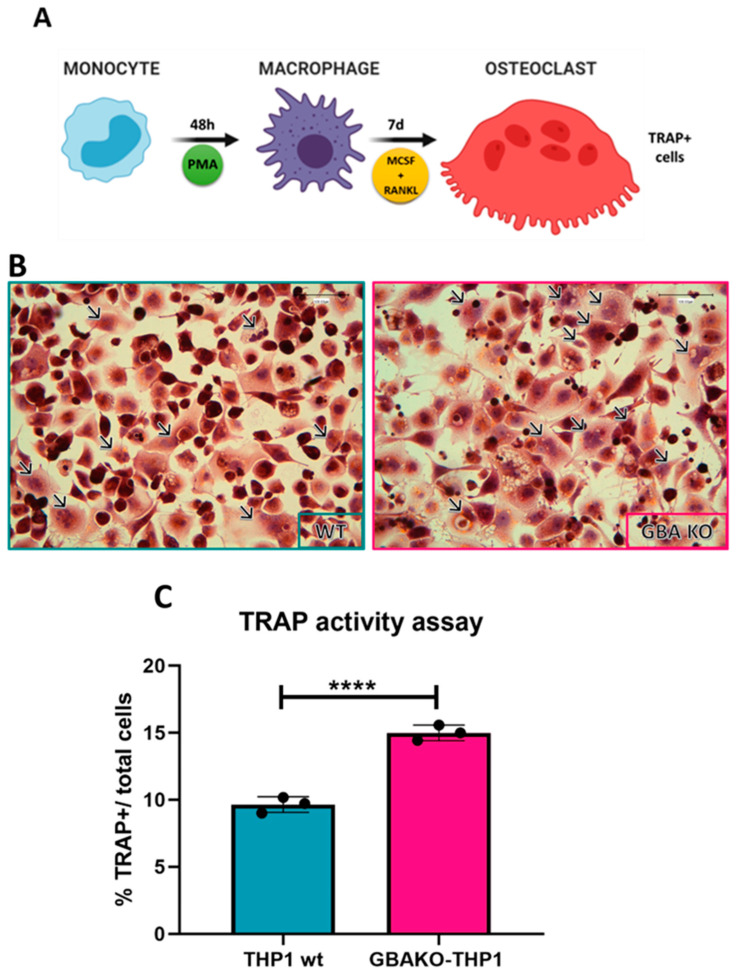
Osteoclastogenic differentiation assay in THP-1 wt and GBAKO-THP1 monocytes. (**A**) Schematic representation of the osteoclast differentiation assay. After inducing macrophage differentiation by incubating cells for 48 h with PMA, cells were cultured in the presence of M-CSF and RANKL to induce osteoclastogenesis. After 7 days, TRAP staining was used to identify the osteoclast-like cells. (**B**) TRAP staining (brown) and hematoxylin (purple) staining for osteoclasts and nuclei identification, respectively. Arrows indicate osteoclasts (i.e., TRAP+ cells with 3 or more nuclei); scale bar: 100 µm. (**C**) Quantification of generated osteoclasts expressed as the percentage of the total number of cells that tested TRAP-positive. Data are shown as mean ± SD of three independent experiments. **** *p* < 0.0001 *t*-test. Abbreviations: PMA: phorbol-12 myristate-13 acetate; M-CSF: recombinant human macrophage colony-stimulating factor; RANKL: recombinant human sRANK ligand; TRAP: tartrate-resistant acid phosphatase.

**Figure 2 ijms-24-11204-f002:**
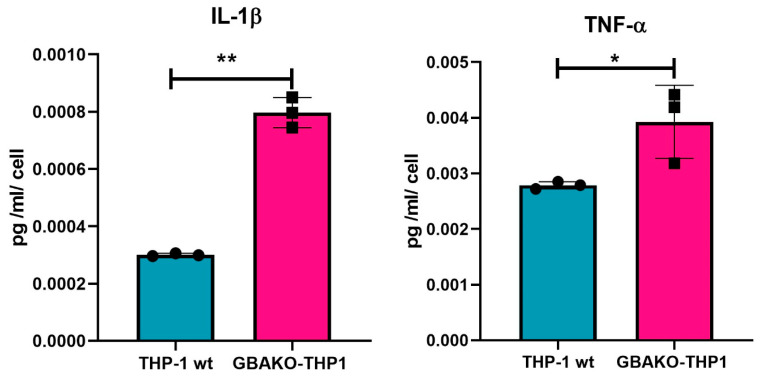
IL-1β and TNF-α levels in the culture supernatant of THP-1 wt and GBAKO-THP1 monocytes. The levels of IL-1β and TNF-α released to the culture media of THP-1 wt and GBAKO-THP1 cells were quantified using a Simple Plex assay (ELLA). Data were normalized via the number of cultured cells and expressed as means ± SD of three independent experiments. * *p* < 0.1 ** *p* < 0.01 *t*-test.

**Figure 3 ijms-24-11204-f003:**
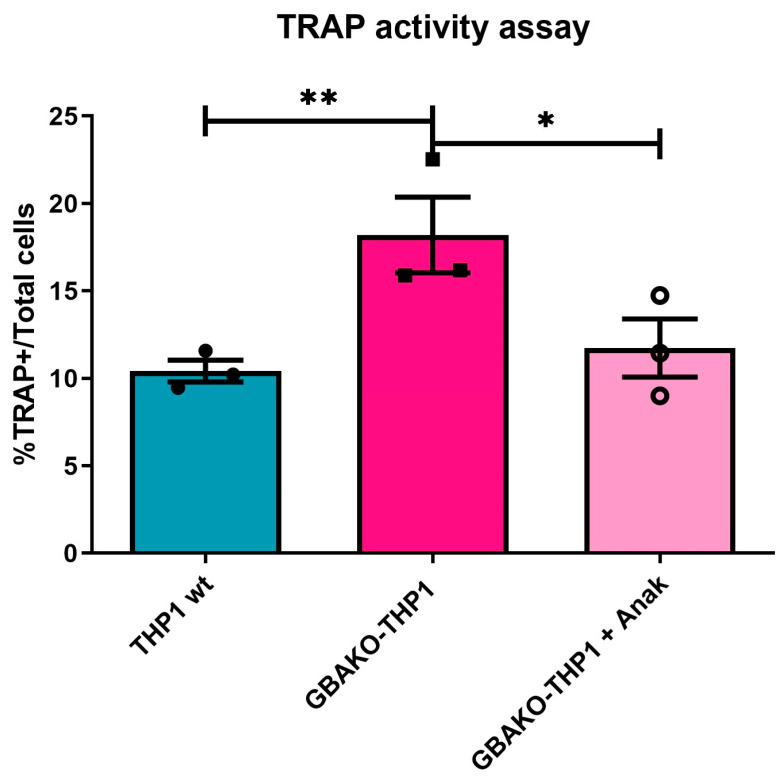
Effect of IL-1 β antagonist on the osteoclastogenic potential of GBAKO-THP1 monocytes. GBAKO-THP1 monocytes were differentiated to osteoclasts in the presence or absence of the IL-1 β receptor antagonist Anakinra. The osteoclasts generated were identified as TRAP-positive cells and quantified. The results were expressed as the percentage of the total number of cells that tested TRAP-positive. Data are shown as mean ± SD of three independent experiments. * *p* < 0.1 ** *p* < 0.01 *t*-test.

**Figure 4 ijms-24-11204-f004:**
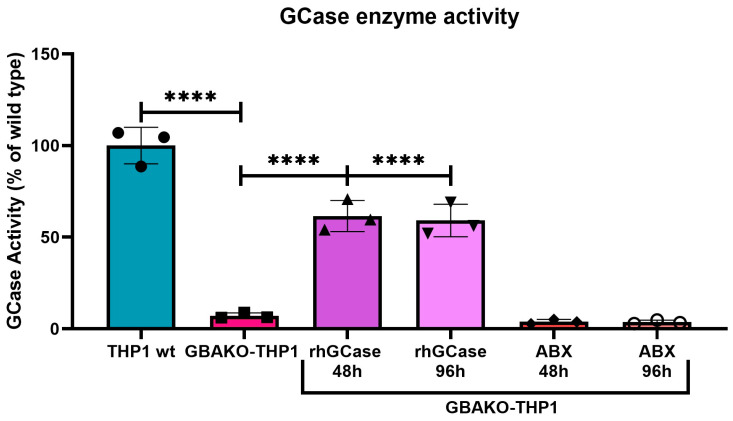
Effect of treatments on GCase activity in GBAKO-THP1 monocytes. GBAKO-THP1 cells were treated with 1.6 µM of rhGCase or 100 µM of ABX. After 48 or 96 h, the GCase activity was assessed. Only rhGCase treatment resulted in increased GCase activity, while ABX did not show any effect. Data are expressed as the percentage of the GCase activity detected in wild-type cells and are shown as mean ± SD of three independent experiments. **** *p* < 0.0001 one-way ANOVA.

**Figure 5 ijms-24-11204-f005:**
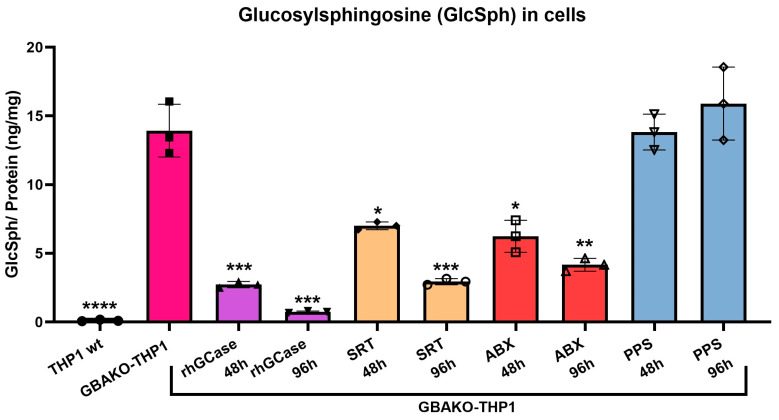
Effect of different treatments on intracellular GlcSph levels of GBAKO-THP1 monocytes. GBAKO-THP1 cells were treated with rhGCase (1.6 µM), D, L-threo-PDMP (as SRT, 20 µM), ABX (100 µM), and PPS (5 μg/mL) for 48 and 96 h. Intracellular GlcSph was measured using LC-MS/MS. rhGCase, SRT, and ABX treatments significantly reduced intracellular GlcSph levels, but PPS had no effect. Data were normalized by the intracellular protein amount and shown as mean ± SD of three independent experiments. * *p* < 0.1 ** *p* < 0.01 *** *p* < 0.001 **** *p* < 0.0001 one-way ANOVA.

**Figure 6 ijms-24-11204-f006:**
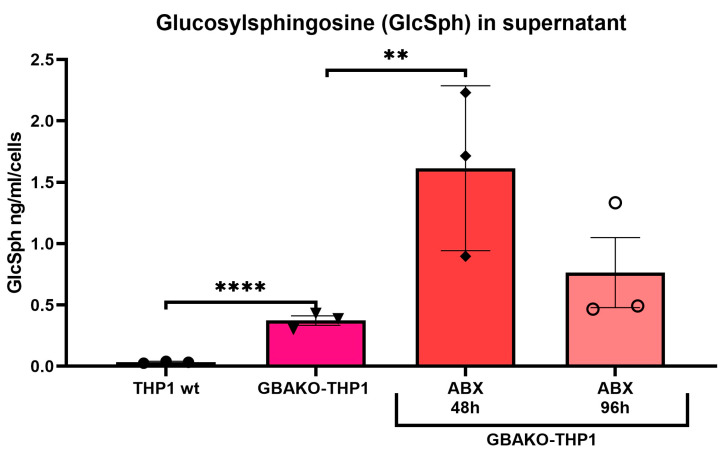
Effect of Ambroxol on extracellular GlcSph levels of GBAKO-THP1 monocytes. Cells were treated with 100 µM of ABX for 48 and 96 h, and the GlcSph levels in the culture media were measured using LC-MS/MS. Ambroxol treatment increased GlSph release to the culture media. GlcSph levels were normalized by the number of cultured cells, and data are expressed as mean ± SD of three independent experiments. ** *p* < 0.01 **** *p* < 0.0001 one-way ANOVA.

**Figure 7 ijms-24-11204-f007:**
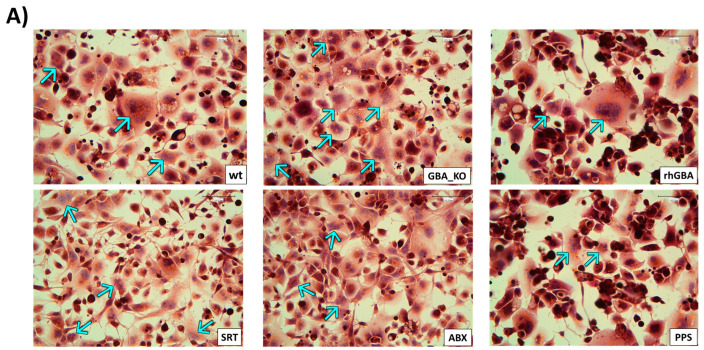
Effect of the different treatments on GBAKO-THP1 monocyte osteoclastogenic potential. GBAKO-THP1 cells were treated using rhGCase (1.6 µM), D, L-threo-PDMP (as SRT, 20 µM), ABX (100 µM), and PPS (5 μg/mL) throughout the osteoclastogenic differentiation process. TRAP assay was used to identify the osteoclast-like cells obtained after 7 days. (**A**) Representative TRAP staining (brown) and hematoxylin (purple) staining for osteoclasts and nuclei identification, respectively. Arrows indicate osteoclasts (i.e., TRAP+ cells with 3 or more nuclei); scale bar: 100 µm. (**B**) All treatments induced a reduction in the number of osteoclast-like cells differentiated from the GBAKO-THP1 cells. The results are expressed as the percentage of total cells that resulted in TRAP-positive at the end of the differentiation protocol. Data are shown as mean ± SD of three independent experiments. * *p* < 0.1 *** *p* < 0.001 **** *p* < 0.0001 one-way ANOVA.

**Figure 8 ijms-24-11204-f008:**
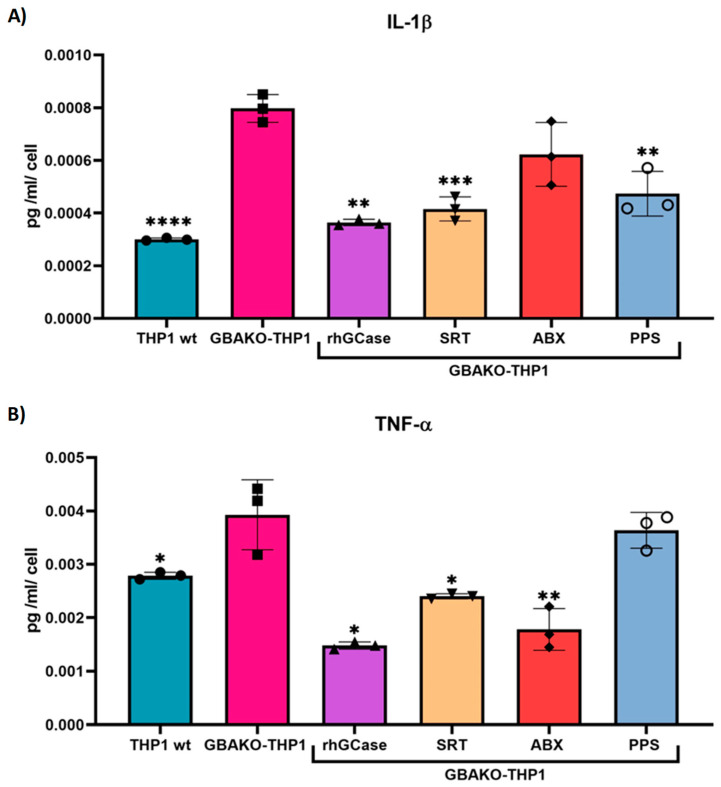
Effect of the different treatments on IL-1β and TNF-α release in GBAKO-THP1 monocytes. GBAKO-THP1 cells were treated with rhGCase (1.6 µM), D, L-threo-PDMP (as SRT, 20 µM), ABX (100 µM), and PPS (5 μg/mL) for 48 h. IL-1β (**A**) and TNF-α (**B**) were assessed in the supernatant of cultured cells using Simple Plex assay (ELLA). Results were normalized via the number of cultured cells. Treated GBAKO-THP1 showed decreased levels of one or both cytokines in comparison with untreated GBAKO-THP1. Data are shown as mean ± SD of three independent experiments. * *p* < 0.1 ** *p* < 0.01 *** *p* < 0.001 **** *p* < 0.0001 *t*-test.

**Figure 9 ijms-24-11204-f009:**
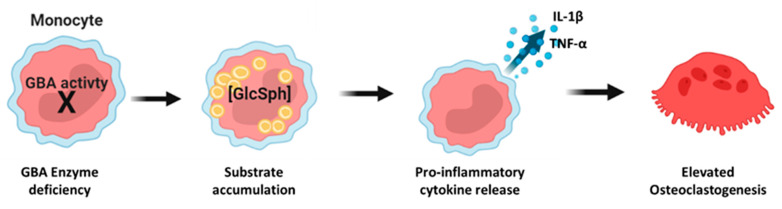
Schematic pathophysiology in the new monocyte GD model.

## Data Availability

Data available on request due to restrictions e.g., privacy or ethical.

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
