# Peer review of "Exploring the Pathophysiologic Cascade Leading to Osteoclastogenic Activation in Gaucher Disease Monocytes Generated via CRISPR/Cas9 Technology"

_ijms, 2023, doi:10.3390/ijms241311204_

Round 1
Reviewer 1 Report
The article by Ormazabal et al "Exploring the pathophysiologic cascade leading to osteoclastogenic activation in Gaucher Disease monocytes generated by 3CRISPR/Cas9 technology" covers a potentially interesting and emerging topic related to the Gaucher Disease pathogenesis. In this sense, this remains to be potentially interesting for the IJMS readers. I regard the main point of this paper as highly attractive as well as the results are clearly presented. The text does not contain any major errors, therefore I have some minor comments and recommendations:
1. There is a need to provide slightly more expanded introduction shortly
mentioning/describing pharmacoeconomical aspects of Gaucher disease and its impact of modern healthcare.
2. "According to the manufacturer’s instructions" - that sentence should be replaced by method describe
3. Following references should be added and properly cited within the main text to improve the quality of manuscript
- Mela A, Poniatowski ŁA, Drop B, Furtak-Niczyporuk M, Jaroszyński J, Wrona W, Staniszewska A, Dąbrowski J, Czajka A, Jagielska B, Wojciechowska M, Niewada M. Overview and Analysis of the Cost of Drug Programs in Poland: Public Payer Expenditures and Coverage of Cancer and Non-Neoplastic Diseases Related Drug Therapies from 2015-2018 Years. Front Pharmacol. 2020 Aug 14;11:1123. doi: 10.3389/fphar.2020.01123.
- Peng Y, Liou B, Lin Y, Mayhew CN, Fleming SM, Sun Y. iPSC-derived neural precursor cells engineering GBA1 recovers acid β-glucosidase deficiency and diminishes α-synuclein and neuropathology. Mol Ther Methods Clin Dev. 2023 Mar 15;29:185-201. doi: 10.1016/j.omtm.2023.03.007.
- Kubaszewski Ł, Wojdasiewicz P, Rożek M, Słowińska IE, Romanowska-Próchnicka K, Słowiński R, Poniatowski ŁA, Gasik R. Syndromes with chronic non-bacterial osteomyelitis in the spine. Reumatologia. 2015;53(6):328-36. doi: 10.5114/reum.2015.57639.
- Kumar M, Srikanth MP, Deleidi M, Hallett PJ, Isacson O, Feldman RA. Acid ceramidase involved in pathogenic cascade leading to accumulation of α-synuclein in iPSC model of GBA1-associated Parkinson's disease. Hum Mol Genet. 2023 May 18;32(11):1888-1900. doi: 10.1093/hmg/ddad025. PMID: 36752535; PMCID: PMC10196677.
4. In some places the use of English throughout the whole manuscript could be improved on.
Completing this gaps will have an impact on the understanding the aim of the study and, from my point of view, is absolutely necessary.
minor review
Author Response
We thank the reviewers for their comments and the suggestions that have been incorporated in the revised version of the manuscript.
Response to reviewer’s comments
REVIEWER 1:
The article by Ormazabal et al "Exploring the pathophysiologic cascade leading to osteoclastogenic activation in Gaucher Disease monocytes generated by 3CRISPR/Cas9 technology" covers a potentially interesting and emerging topic related to the Gaucher Disease pathogenesis. In this sense, this remains to be potentially interesting for the IJMS readers. I regard the main point of this paper as highly attractive as well as the results are clearly presented. The text does not contain any major errors, therefore I have some minor comments and recommendations:
We thank the reviewer for this positive comment
- There is a need to provide slightly more expanded introduction shortly
mentioning/describing pharmacoeconomical aspects of Gaucher disease and its impact of modern healthcare.
A short statement on the impact of ERT and SRT therapies on the health system has been added in the introduction section of the manuscript as requested.
- "According to the manufacturer’s instructions" - that sentence should be replaced by method describe.
The sentence "According to the manufacturer’s instructions” was replaced by a detailed description of the methods
- Following references should be added and properly cited within the main text to improve the quality of manuscript
- Mela A, Poniatowski ŁA, Drop B, Furtak-Niczyporuk M, Jaroszyński J, Wrona W, Staniszewska A, Dąbrowski J, Czajka A, Jagielska B, Wojciechowska M, Niewada M. Overview and Analysis of the Cost of Drug Programs in Poland: Public Payer Expenditures and Coverage of Cancer and Non-Neoplastic Diseases Related Drug Therapies from 2015-2018 Years. Front Pharmacol. 2020 Aug 14;11:1123. doi: 10.3389/fphar.2020.01123.
- Peng Y, Liou B, Lin Y, Mayhew CN, Fleming SM, Sun Y. iPSC-derived neural precursor cells engineering GBA1 recovers acid β-glucosidase deficiency and diminishes α-synuclein and neuropathology. Mol Ther Methods Clin Dev. 2023 Mar 15;29:185-201. doi: 10.1016/j.omtm.2023.03.007.
- Kubaszewski Ł, Wojdasiewicz P, Rożek M, Słowińska IE, Romanowska-Próchnicka K, Słowiński R, Poniatowski ŁA, Gasik R. Syndromes with chronic non-bacterial osteomyelitis in the spine. Reumatologia. 2015;53(6):328-36. doi: 10.5114/reum.2015.57639.
- Kumar M, Srikanth MP, Deleidi M, Hallett PJ, Isacson O, Feldman RA. Acid ceramidase involved in pathogenic cascade leading to accumulation of α-synuclein in iPSC model of GBA1-associated Parkinson's disease. Hum Mol Genet. 2023 May 18;32(11):1888-1900. doi: 10.1093/hmg/ddad025. PMID: 36752535; PMCID: PMC10196677.
References from Mela et al.; Peng et al. and, Kumar et al., have been included in the revised version of the manuscript. However, the subject of the paper by Kubaszewski et al., seems to be out of the scope of this manuscript. If the reviewer still thinks that the reference is missing, we kindly ask to point out to what the section of the text he/she refers to.
In some places the use of English throughout the whole manuscript could be improved on.
The English grammar has been revised by a native English speaker.
Completing this gaps will have an impact on the understanding the aim of the study and, from my point of view, is absolutely necessary.
Reviewer 2 Report
Attached

Minor editing of English language required
Author Response
We thank the reviewers for their comments and the suggestions that have been incorporated in the revised version of the manuscript.
Response to reviewer’s comments
REVIEWER 2:
- Please provide reference for line 58-61.
A reference has been provided as requested
- Please add another 2-3 latest references from the last 5 years along with your current reference number 6 for bone remodelling. There is plenty out there on this topic.
Three additional recently published references (11-13 in the revised version of the manuscript) regarding bone remodeling have been included.
- Please mention ‘PBMCs’ in line 88 in full and short form rather than in line 113.
PBMCs is now mentioned in full and short form in line 88 (line 93 in the revised version).
- Line 103 – please remove ‘given the advantages offered by this model’, as it is not really an advantage, it is more of a convenience – which is understandable in terms of the amount of work needed for this manuscript.
The phrase: “given the advantages offered by this model” has been replaced by “given the convenience of this model”.
- Any particular reason why IL-6 was not investigated along with the other two inflammatory cytokines? You have mentioned it in your introduction and it is important that you investigate it considering its prevalence in these conditions. Please add data for IL-6 as well for figure 2 as well as for figure 8.
IL-6 levels were measured along with TNF-α and IL-1β. However, no differences were detected between the GBAKO and WT cells. This data has been added to the main text of the manuscript.
- I notice that all your data is represented as mean + SD. Was your data checked for normal distribution? If yes- then mean is fine – if not, please double check and change to median if the data were found to be non-parametric.
As requested, the normal distribution of the data has been confirmed using the Shapiro-Wilk normality test. This information is now included in the methods section (statistical analysis) of the revised version of the manuscript
- Please add a section after discussion titled ‘Limitations and conclusions’ to critique your current work and summarise your results. In terms of limitations: One of the major limitations of your study is that while you justify using cell line which is more isogenic in comparison to patient derived cells and are ‘convenient’ to work with – patient derived cells are representative of the variability that actually exists in reality. Especially, if you want a realistic model for drug screening- it is better that a system indicates variation at early stages which is likely with patient derived models; rather than a drug passing in earlier stage of a cell line model and then failing in the next steps. So while this model may have been developed well, its highly likely that even if drug screen past this model – it may not progress towards drug development. Please reflect on this and critique your work in the limitations. There is plenty evidence in literature outlining why cell lines are not truly representative of conditions in vivo and why ex vivo patient cellular models better reflect human physiology.
We thank the reviewer for this comment. As requested, a section entitled “Limitations and conclusions” has been included in the revised version of the manuscript
We agree with the reviewer on the fact that our model do not represent the variability that indeed exist in nature and in particular among GD patients; therefore, the results obtained using it might not apply to all GD patients. This is the main drawback of the model and this point has been discussed in the new version of the manuscript. Besides, the statement (in Introduction section) pointing out the variability of patient’s derived cells as a limitation has been eliminated.
Regarding drug testing, we understand the reviewer’s point. Ideally, a drug selected using in the early stages of the screening, a system that represents variability would have more chances to pass the next steps of development. However, although cells derived from different patients might be used to test the effect of few compounds, this setting is no suitable for high throughput screenings (HTS), since it is not feasible to test the whole set of molecules (usually thousands) in cell lines derived from different patients. This would multiply the cost of screening by the number ofdifferent patient’s cells used.
On the other hand, although we acknowledge the value of patient’s derived cells and in particular, those differentiated from iPSCs, HTS experiments require a high number of cells which it is difficult to obtain using primary cultured cells and even more difficult using cells differentiated from iPSCs (which would be needed to test effects on not accessible target tissues). These limitations make drug HTS in patient’s derived cells extremely difficult, expensive and not cost-effective
For these reasons, we still think that our system would be much better for high throughput screening experiments, recognizing the limitations and acknowledging that the obtained results should be confirmed in systems that consider the variability seen in nature, such us patient’s derived cells.
- For conclusion - summarise your findings in one paragraph.
A paragraph with the main conclusion has been included in the Limitations and conclusions section of the revised manuscript
- In materials and methods, 4.1- please mention where you got the cell line from (ATCC/ another company)?
The requested information has been added to the materials and methods section (5.1 in the revised version)
- Please check the font for line 425.
The font was corrected
- You have provided supplementary materials in support of your manuscript. However, they have not been cited in your manuscript. So, as a reader – I have no clue at what point in your manuscript would you want me to look at your supplementary data. Please clarify.
We apologize for this mistake; the figure is now cited in the main text of the revised version of the manuscript.
- Finally – I think there are at least 9 self-cited articles (if not more) out of a total of 42. Ideally, it is advised to not exceed 15% of references for self-citation, which would be a maximum of 6 references from a total of 42. Kindly rectify this.
Indeed, we overlooked this point. The number of self-cited articles has been reduced to 7 from a total of 50.
Round 2
Reviewer 2 Report
Exploring the pathophysiologic cascade leading to osteoclastogenic activation in Gaucher Disease monocytes generated by CRISPR/Cas9 technology – round 2
Suggestion: Accept in present form – needs minor language check
The authors have addressed all of my queries and have responded to all of them giving their justification. I am happy for this paper to be published after minor spell check by the authors and wish them all the best.
Minor editing of English language required
Author Response
We thank the reviewer again for their comments and suggestion of checking the language speck, which has been revised in the new version of the manuscript.